# Retrieval Augmented Zero-Shot Enzyme Generation for Specified Substrate

**Jiahe Du**[1]  **Kaixiong Zhou**[2]  **Xinyu Hong**[3]  **Zhaozhuo Xu**[4]  **Jinbo Xu**[3]  **Xiao Huang**[1]

## Abstract

Generating novel enzymes for target molecules in zero-shot scenarios is a fundamental challenge in biomaterial synthesis and chemical production. Without known enzymes for a target molecule, training generative models becomes difficult due to the lack of direct supervision. To address this, we propose a retrieval-augmented generation method that uses existing enzyme-substrate data to guide enzyme design. Our method retrieves enzymes with substrates that share structural similarities with the target molecule, leveraging functional similarities in catalytic activity. Since none of the retrieved enzymes directly catalyze the target molecule, we use a conditioned discrete diffusion model to generate new enzymes based on the retrieved examples. An enzyme-substrate relationship classifier guides the generation process to ensure optimal protein sequence distributions. We evaluate our model on enzyme design tasks with diverse real-world substrates and show that it outperforms existing protein generation methods in catalytic capability, foldability, and docking accuracy. Additionally, we define the zero-shot substrate-specified enzyme generation task and introduce a dataset with evaluation benchmarks.

## 1. Introduction

Substrate-specified enzyme generation aims to design new proteins that catalyze reactions to specific new molecules and benefits a wide array of scientific fields, including biomaterials synthesis and chemical production innovation (Meghwanshi et al., 2020; Robinson, 2015; Jegannathan

---

[1]Department of Computing, The Hong Kong Polytechnic University, Hung Hom, Hong Kong SAR, China [2]Department of Electrical and Computer Engineering, North Carolina State University, Raleigh, NC, USA [3]Beijing MoleculeMind Co.,Ltd., Beijing, China [4]Department of Computer Science, Stevens Institute of Technology, Hoboken, NJ, USA. Correspondence to: Xiao Huang <xiao.huang@polyu.edu.hk>.

*Proceedings of the 42nd International Conference on Machine Learning*, Vancouver, Canada. PMLR 267, 2025. Copyright 2025 by the author(s).

& Nielsen, 2013; Paraschiv et al., 2022; Nam et al., 2024). Taking the artificial compound of 1,2,3-trichloropropane (TCP) as an example, it is extensively utilized as a chemical intermediate and solvent despite its toxicity and resistance to biodegradation (Agency for Toxic Substances and Disease Registry, 2021; Cheremisinoff & Rosenfeld, 2011), which leads to persistent groundwater contaminant. Researchers are actively engaged in discovering or engineering enzymes capable of biodegrading TCP (Bogale et al., 2020; Samin & Janssen, 2012). Since there is no existing natural enzymes for TCP, the synthesis paradigm only relies on the expertise of replicating molecular structure of other natural heme-proteins (Zambrano et al., 2022) and lacks the efficiency to discover novel and effective enzymes for the specific substrate.

The recent emergence of deep learning based protein generation shows great potential for enzyme design due to their unprecedented accuracy in structure and function prediction. A portion of these methods falls under the category of unconditional generation, such as ProGen2 (Nijkamp et al., 2023) and ProtGPT2 (Ferruz et al., 2022), possessing the capability to generate protein sequences that fold into stable and functional structures and resemble real proteins, without relying on the predefined substrate. The other subset of these methods is characterized by conditional generation, consisting of ligand-conditioned sequence design and structure generation. The ligand-conditioned sequence design models (Gruver et al., 2023; Martinkus et al., 2023) are proposed to synthesize therapeutic antibodies treating well to the antigen ligands. On the other hand, the ligand-conditioned structure generation methods, like LigandMPNN (Dauparas et al., 2025) and RFdiffusionAA (Krishna et al., 2024), generate proteins structurally docking to a given target. By ensuring spatial compatibility, these methods generate effective proteins associated with enhanced biological function and stability in complex cellular environments.

While the unconditional approaches fail to match requirements, existing work of conditional generation cannot be repurposed directly to generate desired enzymes that catalyze specific substrates represented as small molecules. Particularly, the ligand condition of these models is amino acid sequences of antigens but our substrates are small molecules. The enzyme substrates exhibit a vast chemical space with high structural diversity, including variations in functional

groups, stereochemistry, and electronic properties, which make it challenging to learn the interactions with enzymes. In addition, the catalytic capability of an enzyme is not solely determined by how it structurally interacts with the substrate molecule, so these models are not yet capable of synthesizing functional enzymes. The label-conditioned generative method, i.e. ZymCTRL (Munsamy et al., 2022), takes an Enzyme Commission (EC) number and outputs a corresponding enzyme sequence. It requires prior knowledge about the expected enzyme's EC, which relays human expertise heavily.

In this study, we formally define the task of zero-shot substrate-specified enzyme generation and identify two primary challenges associated with it. The first challenge is the complete absence of positive samples. For instance, without any effective enzymes for TCP as training data, it is difficult to train or fine-tune a model to generate enzymes that catalyze TCP. A potential solution to this challenge is the Retrieval-Augmented Generation (RAG). Specifically, RAG-based methods sample protein sequences as prompts and subsequently instruct models to generate sequences that are structurally and/or functionally similar (Ma et al., 2024; Alamdari et al., 2023; Lewis et al., 2020). However, the problem of retrieving proteins without relying on an exemplar enzyme needs to be addressed, as the only input is the target substrate. The second challenge is the generation of proteins that diverge from training data. The generated TCP enzyme must differ from recorded enzymes, as none in the record can effectively catalyze TCP molecules. This divergence requirement extends to enzymes for other new substrates. Since the mainstream training methods focus on recovering recorded data, a new approach is required—one that trains models to generate enzymes that are both divergent from existing records and capable of catalyzing different target molecules. Furthermore, there is currently no comprehensive evaluation framework for zero-shot enzyme generations. While Johnson et al. (2025) and Song et al. (2024a) introduced certain metrics for computationally designed enzymes, there is a lack of refined datasets for zero-shot settings and multiple-perspective evaluations, as the substrate-specified enzyme generation task has not yet been fully formulated.

To address these two challenges, we propose **S**ubstrate-specified **enz**yme generator (SENZ). Our main contributions are as follows:

- We formally define the **task** of substrate-specified enzyme generation and present a curated dataset. This dataset consists of the substrate-enzyme pairs that are extracted from the known enzymes. We further partition it into training and test subsets without overlap in terms of proteins and small molecules to secure the zero-shot setting.

- We propose a substrate-indexed **retrieval** method to search the functionally-similar enzymes as prompting signals. The key merit is the enzymes associated with the structurally close substrates exhibit similar catalyzing properties. Considering a query substrate, we compare the structural closeness with other stored molecules and retrieve the pairwise enzyme data of top-ranking molecules. This approach is distinct from traditional protein retrieval since it retrieves based on substrate similarity instead of protein similarity, as traditional protein retrieval does.

- We employ a discrete diffusion model to generate new enzymes based on the retrieved ones and utilize a substrate-enzyme catalyzing classifier as **guidance** for the generative process. The classifier transforms the complicated catalytic relationship into a continuous and differentiable function for optimizing the generator. With different substrates, it guides the generation toward different directions distinct from the whole record data distribution.

- Experimental results in designing enzymes for particular substrates demonstrate that our model can generate novel enzymes of superior quality. Compared with rule-, unconditioned-, sequence-, and structure-based methods, our framework generates proteins showing high enzymatic capability and high foldability.

## 2. Substrate-Specified Enzyme Generation Task

We define the substrate-specified enzyme generation task by specifying the model's input and output, along with the training and testing data and evaluation methods.

**Problem definition.** The task involves generating a protein that serves as the enzyme for the target molecule. Let $\mathbf{m}$ denote the Simplified Molecular-Input Line-Entry System (SMILES) (Weininger, 1988) string representation of the molecule and let $\mathbf{x}$ denote the protein sequence. We have $\mathbf{x} = (a_1, a_2, a_3, ..., a_l) \in \mathbb{A}^l$ where $a_i$ is an amino acid and $\mathbb{A}$ is the vocabulary of amino acids together with related tokens including gap ("-"). Henceforth, the amino acid $a$ can be represented as a one-hot vector, and we do not differentiate between the protein sequence and the sequence of one-hot vectors, which means $\mathbf{x} \in \mathbb{A}^l$ is a matrix with shape $l \times |\mathbb{A}|$. Let $\mathbb{P}$ denote the domain of all protein sequences and let $\mathbb{M}$ denote the domain of all molecular SMILES strings. The function $G : \mathbb{M} \rightarrow \mathbb{P}$ means the task of substrate-specified enzyme generation, which can be defined as $\mathbf{x} = G(\mathbf{m}; \boldsymbol{\theta})$ where $\boldsymbol{\theta}$ is the set of $G$'s parameters. If $G$ is a machine-learning model, the training process is

given by:

$$\boldsymbol{\theta}^* = \arg\min_{\boldsymbol{\theta}} \mathcal{L}(G(\mathbf{m}_{\mathcal{D}}; \boldsymbol{\theta}), \mathbf{x}_{\mathcal{D}}, \mathbf{m}_{\mathcal{D}}). \tag{1}$$

$\mathbf{x}_{\mathcal{D}}$ and $\mathbf{m}_{\mathcal{D}}$ are the enzyme and molecule in training set $\mathcal{D}$, respectively, $\boldsymbol{\theta}^*$ is the optimal parameters, and $\mathcal{L}$ is the loss function. The input can include various types of data: Enzyme Commission (EC) label of string $s_{\mathrm{EC}} = N_1.N_2.N_3.N_4$, three-dimensional conformation structure of the target substrate $\mathbf{C}_{\mathbf{m}}$ or an existing enzyme $\mathbf{C}_{\mathbf{x}}$. The generative function can be extended as below:

$$\mathbf{x} = G(\mathbf{m}, s_{\mathrm{EC}}, \mathbf{C}_{\mathbf{m}}, \mathbf{C}_{\mathbf{x}}), \tag{2}$$

where $s_{\mathrm{EC}}, \mathbf{C}_{\mathbf{m}}, \mathbf{C}_{\mathbf{x}}$ are all optional input parameters for $G(\cdot)$, but $\mathbf{m}$ is the required input.

**Data construction.** For this task, we construct a dataset of substrate-enzyme pairwise relationships extracted from public raw data, as illustrated in Fig. 1(a). Each record in raw data comprises the SMILES representations of a chemical reaction with a specific enzyme. To identify the specific substrate in each chemical reaction, we select the least common reactant among all reactants in the database, treating it as the specific substrate for the enzymes involved in that reaction. This approach is grounded in the established observation of substrate specificity (Jackson et al., 2010). Consequently, we define the "substrate-enzyme" relation $(\mathbf{m}, \mathbf{x})$ as protein $\mathbf{x}$ being the enzyme of molecule $\mathbf{m}$, and the training dataset $\mathcal{D}$ can be defined as follows:

$$\mathcal{D} = \{(\mathbf{m}, \mathbf{x})\}, \ \mathbf{x} \text{ is the enzyme of } \mathbf{m}. \tag{3}$$

The substrate-enzyme pair $(\mathbf{m}, \mathbf{x})$ is the element of the dataset as in Eq. (3).

**Zero-shot data split.** All substrate-enzyme pairs $(\mathbf{m}, \mathbf{x})$ are split into $\mathcal{D}$ for training, $\mathcal{D}_{\mathrm{valid}}$ for validation and $\mathcal{D}_{\mathrm{test}}$ for testing. To avoid of data leakage, two rules are designed for any two $(\mathbf{m}_1, \mathbf{x}_1)$ and $(\mathbf{m}_2, \mathbf{x}_2)$ in different subsets: 1. *Molecules from different subsets should not be the same*, i.e. $\mathbf{m}_1 \neq \mathbf{m}_2$; 2. *Any two protein sequences from different subsets, i.e., $\mathbf{x}_1$ and $\mathbf{x}_2$, should not have an overlap of more than 30% (with an identity exceeding 30%)*. The split forms a zero-shot setting. Take the target molecule TCP as an example. TCP is in $\mathcal{D}_{\mathrm{test}}$ and the model $G$ is generating enzyme for TCP. $G$ has never trained with TCP because TCP is not in $\mathcal{D}$. $G$ has never seen proteins similar to TCP's ground truth enzymes because all of them are only in $\mathcal{D}_{\mathrm{test}}$, and all proteins in $\mathcal{D}$ have at least 70% different from them. Therefore generating enzyme for TCP and any molecules in $\mathcal{D}_{\mathrm{test}}$ is zero-shot.

**Evaluation.** Regardless of the input data, models should be evaluated using consistent metrics. An evaluation model $f_{\mathrm{eval}}$ scores the generated protein $\mathbf{x}$ as follows:

$$y = f_{\mathrm{eval}}(\mathbf{x}, \mathbf{m}), \tag{4}$$

where $\mathbf{m}$ is optional. If the evaluation focuses solely on the generated protein, $\mathbf{m}$ is not required. Given different functions of $f_{\mathrm{eval}}$, the ideal training process should be framed as a multi-objective optimization problem. However, in the substrate-specified enzyme generation task, we prioritize catalytic capability above all and thus focus primarily on the corresponding $f_{\mathrm{eval}}$.

## 3. Substrate-Specified Enzyme Generator

We present **S**ubstrate-specified **enz**yme generator (SENZ), a novel approach designed to **retrieve** enzymes based on a new target substrate and subsequently **generate** new enzymes from the retrieved ones with the help of a **guidance** training method.

### 3.1. Substrate-Indexed Enzyme Retrieval Module

Since there are no existing enzymes for a target substrate in the zero-shot generation setting, it is crucial to retrieve the related data record without relying on the ground truth enzyme sequence as an anchor. In order to retrieve a set of related proteins $\mathbb{P}^{(\mathbf{m})}$ for the target molecule $\mathbf{m}$, **a relational database** is constructed and **a substrate-similarity based retrieval rule** is designed. The superiority is demonstrated by only querying with molecule $\mathbf{m}$, while traditional protein retrieval methods require an anchor sequence to search for similar sequences.

**Substrate-enzyme relational database.** We adopt training set $\mathcal{D}$ in Eq. (3) as a relational database of substrate-enzyme pairs $(\mathbf{m}, \mathbf{x})$. $\mathcal{D}$ contains substrate-indexed enzymes, in which substrates are non-unique indices for corresponding protein sequences as shown in Fig. 1(b) green part.

**Retrieval by substrate-similarity.** Based on relational database $\mathcal{D}$, we then retrieve enzymes whose substrates exhibit high similarity to the target molecule, with the expectation that the generated enzyme will incorporate beneficial features from the retrieved ones. This approach is based on the observation that enzymes catalyzing highly similar substrates may also share some similarities (Goldman et al., 2022). We denote all molecules in $\mathcal{D}$ as set $\mathcal{D}_{\mathbf{m}}$. Querying $\mathcal{D}$ with a molecule $\mathbf{m}$ gets a protein set $\mathbb{P}^{(\mathbf{m})}$ as follows:

$$\mathbb{P}^{(\mathbf{m})} = \begin{cases} \{\mathbf{x} | (\mathbf{m}, \mathbf{x}) \in \mathcal{D}\}, & \mathbf{m} \in \mathcal{D}_{\mathbf{m}}, \quad (5) \\ \displaystyle\bigcup_{i=1}^{d} \mathbb{P}^{(\mathbf{m}_i)} \text{ where } \mathbf{m}_i \in \mathcal{D}_{\mathbf{m}}, & \mathbf{m} \notin \mathcal{D}_{\mathbf{m}}. \quad (6) \end{cases}$$

We consider two cases to retrieve the related enzymes. On one hand, if $\mathbf{m}$ is stored in the relational database as shown in Eq. (5), all protein indexed with $\mathbf{m}$, i.e., $\mathbf{m}$'s enzymes, are obtained by table-checking; otherwise, if $\mathbf{m}$ is not stored ($\mathbf{m} \notin \mathcal{D}_{\mathbf{m}}$) as in Eq. (6), which is the case in the zero-shot enzyme generation, $\mathbb{P}^{(\mathbf{m})}$ consists of a number of $d$ enzymes selected from $\mathcal{D}$ according to following rules.

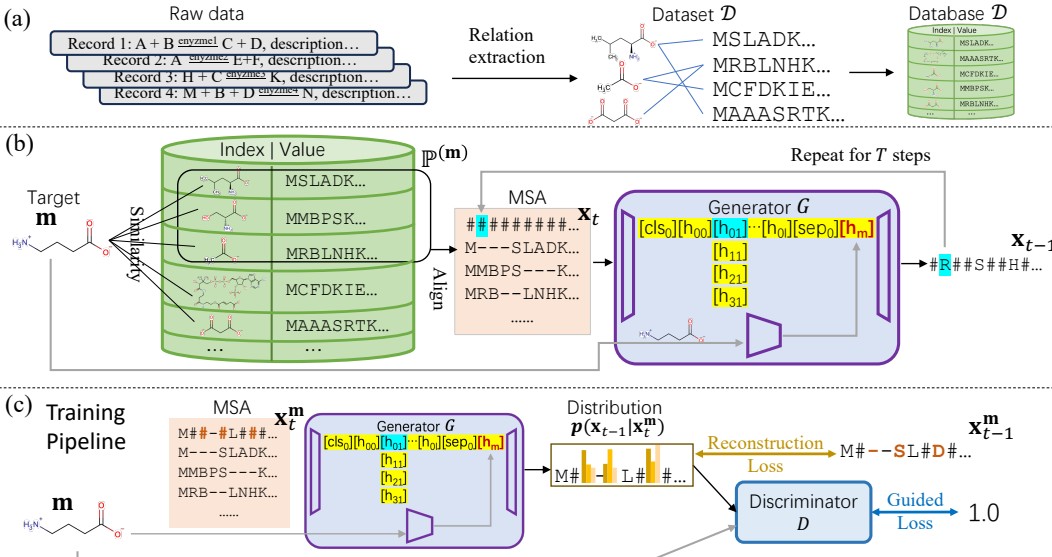

Figure 1: (a) Database extraction. Extracting substrate-enzyme relation from records and constructing a relational database indexing with substrates. (b) Sample pipeline. Retrieve enzymes from the database based on their substrates' similarity to the target molecule. Align them in MSA for the generator and insert a fully masked sequence on top. Predict masks of the top sequence every iteration until the full sequence is unmasked. (c) Training pipeline. A partly masked ground truth enzyme sequence is inserted on top of the retrieved sequences' MSA, and the generator outputs the distribution of amino acids on masked positions. The reconstruction loss measures the distribution difference between the generated and ground truth sequence of one timestep before. The guided loss is the gap between the score of the generated sequence given by a discriminator and the maximum score of 1.

First, all the substrates $\mathbf{m}_i$ in $\mathcal{D}$ are compared with target molecule $\mathbf{m}$ to determine the Tanimoto similarity of their one-hot Morgan fingerprint. The top-$d$ $\mathbf{m}_i$ are selected in descending order based on the similarity to $\mathbf{m}$, represented as $\mathbf{m}_1, ..., \mathbf{m}_d$. Finally a number of $d$ enzymes are gathered from $\mathbb{P}^{(\mathbf{m}_1)}, ..., \mathbb{P}^{(\mathbf{m}_d)}$ to form the retrieval result $\mathbb{P}^{(\mathbf{m})}$.

### 3.2. MSA-based Generator Module

With retrieved enzyme sequences, we transform them into Multiple Sequence Alignment (MSA) format as input and employ **a discrete diffusion model generator** to derive a new enzyme. MSAs are matrices of protein sequences aligned to uniform length through strategic gap insertions, facilitating the comparative analysis of homologous positions across related sequences.

**Discrete noising for enzyme generator.** Our generator $G$, depicted in Fig. 1(b), is an order-agnostic autoregressive diffusion model (Hoogeboom et al., 2022) with an MSA transformer (Rao et al., 2021) backbone. $G$ generates protein sequence by gradually denoising from a fully noised sequence. To begin with, a number of $d$ enzymes within $\mathbb{P}^{(\mathbf{m})}$ are aligned into MSA matrix by ClustalW algorithm (Thompson et al., 1994): $\mathbf{X}^{(\mathbf{m})} = \text{ClustalW}(\mathbb{P}^{(\mathbf{m})}) \in \mathbb{A}^{d \times l}$. A partly noised sequence $\mathbf{x}_t = (a_1, a_2, ..., a_l)$ is inserted on

the top of $\mathbf{X}^{(\mathbf{m})}$ as a new row to formulate data point $\mathbf{X}_t$ at time step $t \leq T$ in the diffusion model:

$$\mathbf{X}_t = \begin{bmatrix} \mathbf{x}_t \\ \mathbf{X}^{(\mathbf{m})} \end{bmatrix} \in \mathbb{A}^{(d+1) \times l}, \text{ and } \sum_{i=1}^{l} \mathbf{1}_{\{a_i = \#\}} = kt. \quad (7)$$

where $a_i = \#$ means position $i$ of $\mathbf{x}_t$ is masked. $\mathbf{1}_{\{a_i = \#\}} = 1$ if $a_i = \#$ otherwise 0. There are $k \cdot t$ masks in $\mathbf{x}_t$. $k$ is the number of increasing masked positions from $\mathbf{x}_t$ to $\mathbf{x}_{t+1}$, so $k \cdot T = l$. Therefore $\mathbf{x}_T = \#^l$ is a totally noised (masked) sequence, and $\mathbf{x}_0$ is the finally generated sequence.

**Discrete denoising at the generative process.** We adopt matrix $\boldsymbol{p} \in [0, 1]^{l \times |\mathbb{A}|}$ to represent the probability of selecting each vocabulary on each position in a length $l$ sequence, and $\boldsymbol{p}(\mathbf{x}_{t-1}|\mathbf{x}_t)$ to represent the conditional probability distribution of $\mathbf{x}_{t-1}$ from unmasking $k$ positions of $\mathbf{x}_t$. Apparently $\mathbf{x}_{t-1} \sim \boldsymbol{p}(\mathbf{x}_{t-1}|\mathbf{x}_t)$ when $\mathbf{x}_t$ is fixed. Our generator $G$ is defined as follows:

$$\mathbf{z} = G(\mathbf{X}_t, \mathbf{m}) = G(\mathbf{x}_t, \mathbf{m}). \quad (8)$$

$$\boldsymbol{p}(\mathbf{x}_{t-1}|\mathbf{x}_t) = \text{softmax}(\mathbf{z}). \quad (9)$$

The Eq. (8)'s second equation holds because $\mathbf{X}_t = [\mathbf{x}_t; \mathbf{X}^{(\mathbf{m})}]$ and $\mathbf{X}^{(\mathbf{m})}$ is decided by $\mathbf{m}$. $\mathbf{z}$ is the model output log-likelihood. Eq. (9) outputs distribution $\boldsymbol{p}(\mathbf{x}_{t-1}|\mathbf{x}_t)$ for sampling by time step. The fully masked sequence $\mathbf{x}_T$ can

be denoised step by step to the final result $\mathbf{x}_0$: $\mathbf{x}_{T-1}$ can be sampled from $\boldsymbol{p}(\mathbf{x}_{T-1}|\mathbf{x}_T)$, and so on $\mathbf{x}_0$ can be sampled from $\boldsymbol{p}(\mathbf{x}_0|\mathbf{x}_1)$. Those are the denoising steps.

Molecule and protein representation fusion: To inject the target substrate $\mathbf{m}$ into the generative learning process, we adopt a learnable molecule encoder (Ahmad et al., 2023). Specifically, a Graph Attention Network (GAT) (Veličković et al., 2018) is used to encode the molecule's graph structure to embedding $h_m$, which has the same shape as token embedding in generative function $G$. $h_m$ is appended at the end of each row in the MSA representation as an additional token, as illustrated in red in Fig. 1(b). This design respects the relative size relationship in terms of atom numbers between an amino acid and the substrate in the real world. Since the MSA transformer in $G$ performs row-wise attention and tied column-wise attention on the MSA matrix, the integration allows $\mathbf{m}$ to influence the generation in $G$ together with the retrieved MSA $\mathbf{X}^{(\mathbf{m})}$.

**Training to mimic distribution.** With ground truth substrate-enzyme pair $(\mathbf{m}, \mathbf{x}^{\mathbf{m}})$ in training set, $G$ output distribution $\boldsymbol{p}(\mathbf{x}_{t-1}|\mathbf{x}_t^{\mathbf{m}}) = \text{softmax}(G(\mathbf{x}_t^{\mathbf{m}}, \mathbf{m}))$ from $\mathbf{x}_t^{\mathbf{m}}$ is trained to consist with training set distribution $\boldsymbol{p}(\mathbf{x}_{t-1}^{\mathbf{m}}|\mathbf{x}_t^{\mathbf{m}})$. Ground truth protein $\mathbf{x}^{\mathbf{m}}$ is the enzyme of molecule $\mathbf{m}$. $\mathbf{x}_t^{\mathbf{m}}$ is partly noised (masked) $\mathbf{x}^{\mathbf{m}}$ at time step $t$ with $kt$ masks. Denoting $P = \boldsymbol{p}(\mathbf{x}_{t-1}^{\mathbf{m}}|\mathbf{x}_t^{\mathbf{m}})$ and $Q = \boldsymbol{p}(\mathbf{x}_{t-1}|\mathbf{x}_t^{\mathbf{m}})$, KL-divergence is used to measure the difference:

$$D_{\text{KL}}(\boldsymbol{p}(\mathbf{x}_{t-1}^{\mathbf{m}}|\mathbf{x}_t^{\mathbf{m}})||\boldsymbol{p}(\mathbf{x}_{t-1}|\mathbf{x}_t^{\mathbf{m}}))$$
$$= D_{\text{KL}}(P||Q) = H(P,Q) - H(P). \quad (10)$$
$$\mathcal{L}_r = H(P,Q) = -\sum_{|\mathbb{A}|} P(i)\log Q(i)$$
$$= CE(\mathbf{x}_{t-1}^{\mathbf{m}}, \text{softmax}(G(\mathbf{x}_t^{\mathbf{m}}, \mathbf{m}))). \quad (11)$$

$D_{\text{KL}}$ is performed on the vocabulary probability dimension of $\boldsymbol{p}$. The second equation in Eq. (11) is derived from $P = \boldsymbol{p}(\mathbf{x}_{t-1}^{\mathbf{m}}|\mathbf{x}_t^{\mathbf{m}}) = \mathbf{x}_{t-1}^{\mathbf{m}}$ and $Q = \boldsymbol{p}(\mathbf{x}_{t-1}|\mathbf{x}_t^{\mathbf{m}}) = \text{softmax}(G(\mathbf{x}_t^{\mathbf{m}}, \mathbf{m}))$. Since $H(P)$ is a constant given $\mathbf{x}^{\mathbf{m}}$, $H(P,Q)$ can measure the difference of our model's distribution to the training set and is adopted as reconstruction loss $\mathcal{L}_r$.

Variable sequence length: Although $\mathbf{x}_{t-1}$ has a fixed length $l$, the represented protein sequence may have a different length. MSA inserts gap tokens ("-") into the origin protein sequence of amino acids to align them. $\mathbf{x}_{t-1}^{\mathbf{m}}$ and $\mathbf{x}_t^{\mathbf{m}}$ are masked from sequence in MSA $\mathbf{X}^{(\mathbf{m})}$, so there are also many "-" in $\mathbf{x}_{t-1}^{\mathbf{m}}$. Based on Eq. (11), $G$ is learned to output the training set sequence distribution $\boldsymbol{p}(\mathbf{x}_{t-1}^{\mathbf{m}}|\mathbf{x}_t^{\mathbf{m}})$. As a result, the probability of "-" can be high in some positions in $G$'s output $\boldsymbol{p}(\mathbf{x}_{t-1}|\mathbf{x}_t^{\mathbf{m}})$, just as the training target $\boldsymbol{p}(\mathbf{x}_{t-1}^{\mathbf{m}}|\mathbf{x}_t^{\mathbf{m}})$. Then "-" will probably be sampled at some position in $\mathbf{x}_{t-1}$. Gaps "-" in the fully sampled sequence $\mathbf{x}_0$ will be removed and thus $\mathbf{x}_0$ is shorter than $l$.

## 3.3. Guided Training Method

We employ **guidance from a catalyzing discriminator** to train generator $G$. The discriminator $D$ evaluates whether a molecule $\mathbf{m}$ and a protein $\mathbf{x}$ are a substrate-enzyme pair with a score $y = D(\mathbf{x}, \mathbf{m})$. $D$ is pre-trained on training set $\mathcal{D}$ and remains frozen during the generator's training.

**Gradient guidance from discriminator.** To generate enzyme $\mathbf{x}$ containing catalytic capability to a molecule $\mathbf{m}$, the frozen $D$ guides the training of $G$ by constructing guided loss $\mathcal{L}_g$ as follow:

$$\mathbf{x}^* = \boldsymbol{p}(\mathbf{x}_{t-1}|\mathbf{x}_t^{\mathbf{m}}) = g(\mathbf{z}), \quad (12)$$
$$y^* = D(\mathbf{x}^*, \mathbf{m}), \quad (13)$$
$$\mathcal{L}_g = 1 - y^*, \quad (14)$$
$$-\partial\mathcal{L}_g/\partial\boldsymbol{\theta}_G = \partial D(\mathbf{x}^*, \mathbf{m})/\partial\boldsymbol{\theta}_G$$
$$= \partial D(\mathbf{x}^*, \mathbf{m})/\partial\mathbf{x}^* \cdot \partial\mathbf{x}^*/\partial\boldsymbol{\theta}_G$$
$$= \nabla_{\mathbf{x}^*}D(\mathbf{x}^*, \mathbf{m}) \cdot \partial\boldsymbol{p}(\mathbf{x}_{t-1}|\mathbf{x}_t^{\mathbf{m}})/\partial\boldsymbol{\theta}_G. \quad (15)$$

$\mathbf{z}$ is the model output log-likelihood in Eq. (8), $g(\cdot)$ is Gumbel-softmax function (Jang et al., 2017) and $\boldsymbol{\theta}_G$ is the parameters of $G$. The gradients derived from the discriminator can be decoupled into three steps: soft protein sequence generation, loss construction, and gradient derivation. First, Eq. (12) transforms the output of $G$ into distribution $\boldsymbol{p}(\mathbf{x}_{t-1}|\mathbf{x}_t^{\mathbf{m}})$ associated with differentiable noises. The $\boldsymbol{p}(\mathbf{x}_{t-1}|\mathbf{x}_t^{\mathbf{m}})$ can be regarded as a "soft" protein sequence, i.e., $\mathbf{x}^*$, at which each token is a continuous amino acid probability instead of one-hot vector. Second, let $y^*$ denote the predicted catalyzing score for $\mathbf{x}^*$ as shown in Eq. (13). We thus construct the guided loss $\mathcal{L}_g$ as the difference between $y^*$ and maximum score 1. By minimizing loss $\mathcal{L}_g$, generator $G$ should be supervised to synthesize soft enzyme sequence $\mathbf{x}^*$ with a score close to 1. Third, when updating generator via $\boldsymbol{\theta}_G \leftarrow \boldsymbol{\theta}_G - \eta \cdot \partial\mathcal{L}_g/\partial\boldsymbol{\theta}_G$, two items needed to be computed according to Eq. (15): $\nabla_{\mathbf{x}^*}D(\mathbf{x}^*, \mathbf{m})$ means the gradient direction of $\mathbf{x}^*$, to which the soft distribution changes can lead to an effective enzyme functioning higher catalyzing probability for target molecule $\mathbf{m}$; $\partial\boldsymbol{p}(\mathbf{x}_{t-1}|\mathbf{x}_t^{\mathbf{m}})/\partial\boldsymbol{\theta}_G$ is the Jacobian matrix describing if the soft sequence changes, how should the parameters within model $G$ correspondingly updates in order to synthesize proteins adhere to the desired distribution of molecule $\mathbf{m}$'s enzymes.

Therefore, both $\mathcal{L}_g$ and $\mathcal{L}_r$ function by providing a changing direction for the output distribution $\boldsymbol{p}(\mathbf{x}_{t-1}|\mathbf{x}_t^{\mathbf{m}})$, except they are for different purposes: *the former one pursues an effective enzyme for $\mathbf{m}$ while the later regularize the generative enzymes to be close to training set $\boldsymbol{p}(\mathbf{x}_{t-1}^{\mathbf{m}}|\mathbf{x}_t^{\mathbf{m}})$.* The final loss $\mathcal{L}$ is the sum of reconstruction loss $\mathcal{L}_r$ from Eq. (11) and the guidance loss $\mathcal{L}_g$ from Eq. (14), expressed as $\mathcal{L} = \mathcal{L}_r + \mathcal{L}_g$, which are used to update the generator.

# 4. Experiment

## 4.1. Dataset for Substrate-Specified Enzyme Generation

Table 1: Enzyme distribution in the split of Enzyme-Substrate Relation Dataset

| dataset | #entry | #mol | #enzyme | #enzyme/mol | | | | #EC |
|---|---|---|---|---|---|---|---|---|
| | | | | 25% | 50% | 75% | max | |
| training | 26757 | 2294 | 8179 | 2 | 4 | 10 | 868 | 819 |
| validation | 4279 | 366 | 2617 | 1 | 3 | 6 | 501 | 381 |
| testing | 3946 | 389 | 2432 | 1 | 2 | 7 | 316 | 553 |
| total | 34982 | 3049 | 13228 | 1 | 4 | 9 | 868 | 1746 |

**We provide a substrate-enzyme relationship dataset** extracted from RHEA[1] database to better evaluate model performance on the substrate-specified enzyme generation task. Statistics of the dataset are shown in Table. 1. The two *rules* in Sec. 2 are strictly followed to avoid data overlap.

## 4.2. Catalytic Activity Evaluation

**Research question: Can SENZ generate proteins with catalytic capability for specified target molecules?** This section compares our model with eight baselines and the ground truth enzymes to evaluate the generated proteins' catalytic capability. Ten sequences are generated in each design task.

**Baselines.** We compare our model with 4 kinds of baselines. The rule-based methods include: a) the ground truth proteins that are recorded to be the enzymes of target molecule; b) randomly generated amino acids sequences as random proteins; c) single position mutation of the ground truth enzymes; and d) the retrieved enzymes based on our substrate-index enzyme retrieval method. The unconditional generation models include ProtGPT2 (Ferruz et al., 2022) and ProGen2 (Nijkamp et al., 2023), which generates protein sequences with a distribution like natural ones while having some distance. The Sequence generation models include: ZymCTRL (Munsamy et al., 2022), which takes an Enzyme Commission (EC) number and outputs a corresponding enzyme; and NOS (Gruver et al., 2023), which is a guided diffusion model for antibody infilling with our modified guided function same as our model for enzyme generation. The structure-based model is LigandMPNN (Dauparas et al., 2025), which refines proteins based on the binding of small molecules.

**Metric.** We adopt the turnover number of the enzyme ($k_{cat}$) to measure its catalytic capability. A well-accepted predictor, UniKP (Yu et al., 2023), is used to predict $\log_{10}(k_{cat})$ value for the generated enzyme on the target molecule. UniKP is trained on the dataset of enzyme-substrate reaction $k_{cat}$.

▷ *Table 2 shows the $\log_{10}(k_{cat})$ of different methods' generated enzymes with targets, from which we observe our model generated proteins have the highest catalytic capability among all.* The predicted $\log_{10}(k_{cat})$ of Ground Truth enzymes are much higher than those of random protein sequences, suggesting the effectiveness of the evaluation metric. Our model generated enzymes have the highest average turnover number among all the compared methods in the designing tasks. The result shows our model is able to generate enzymes with high turnover numbers when evaluated *in silico*. Table 2 also suggests that generated enzymes can outperform Ground Truth natural enzymes, which suggests the natural enzymes are possibly not the most efficient.

## 4.3. Protein Properties Evaluation

**Research question: Can SENZ generate proteins with good quality as well as catalytic capability?** We evaluate the generated sequences for all 389 substrates in the test set with six $f_{eval}$ to validate our model's generated sequence in different protein properties. 10 enzymes are generated for each substrate.

**Metric.** Protein property predictors $f_{eval}$ are adopted in the evaluation, including: a) the predicted local distance difference test (pLDDT) of ESMFold (Lin et al., 2023), which is the confidence score of protein structure prediction in [1, 100]; b) identity with the nearest different known sequence got by BLASTp[2] in SwissProt database[3]; c) the number of clusters with identity over 30%; d) the length of repeat amino acids (Johnson et al., 2025); and e) the successful rate, which quantifies the proportion of successfully generated sequences relative to the total desired number of sequences. Wasserstein distance is used following (Martinkus et al., 2023) in b), and d), and the absolute difference is calculated in c), aiming to describe the distribution difference between the test set and generated enzymes for each target molecule individually.

▷ *Table 3 presents the properties of our method-generated enzymes, highlighting their superior catalytic capability ($\log_{10}(k_{cat})$) and foldability (pLDDT) compared to other neural network methods.* Notably, ZymCTRL exhibits similar properties, but it relies on ground truth EC numbers as input. The process of mapping the target substrate to the correct EC number requires more human expertise than our model. The Wasserstein distance with the test set on BLASTp and the difference in cluster number shows that our model can generate new proteins that have a similar distribution with the test set, suggesting our generated proteins cluster properly to be specific for each target substrate, just like natural enzymes.

---

[1]https://www.rhea-db.org

[2]https://blast.ncbi.nlm.nih.gov/doc/blast-help/downloadblastdata.html

[3]https://ftp.ncbi.nlm.nih.gov/blast/db/swissprot.tar.gz

Table 2: Average $\log_{10}(k_{cat})$ of generated enzymes towards different targets of 7 tasks.

| Type | Model | Sepiap-terin | Propylene oxide | Levo- glu-cosan | cGMP | L-Pro | Pyri-doxine | leukotriene A4(1-) |
|---|---|---|---|---|---|---|---|---|
| Rule | Ground Truth | 0.247 | 0.785 | 0.719 | 0.132 | 0.107 | 0.508 | 0.371 |
| | Random | -0.056 | 0.076 | 0.359 | -0.203 | 0.037 | 0.269 | -0.215 |
| | Mutation | 0.387 | 0.752 | 0.740 | 0.006 | 0.030 | 0.480 | 0.316 |
| | Retrieved | 0.139 | 0.701 | 0.728 | -0.004 | 0.039 | 0.234 | 0.575 |
| Uncond | ProtGPT2 | 0.410 | 0.441 | 0.491 | 0.194 | 0.244 | 0.432 | 0.302 |
| | ProGen2 | 0.234 | 0.423 | 0.529 | 0.410 | 0.385 | 0.517 | 0.351 |
| Sequence | ZymCTRL | -0.091 | 0.444 | 0.505 | 0.174 | 0.109 | 0.549 | 0.268 |
| | NOS | 0.066 | 0.331 | 0.370 | -0.071 | 0.193 | 0.265 | 0.229 |
| Structure | LigandMPNN | 0.125 | 0.641 | 0.707 | 0.079 | 0.358 | 0.333 | 0.429 |
| | Ours | **0.705** | **0.802** | **0.788** | **0.464** | **0.462** | **0.745** | **1.288** |

Table 3: Different properties predicted by their $f_{eval}$ of the generated enzymes for test set.

| Type | Model | $k_{cat}$ ↑ | pLDDT ↑ | WD↓ (BLASTp) | Absolute difference ↓ (#cluster) | WD↓ (#repeat AA) | success rate↑(%) |
|---|---|---|---|---|---|---|---|
| Rule | Test set | 0.363 | - | - | - | - | - |
| | Random | 0.185 | 20.2 | 38.3 | 8.59 | - | - |
| | Mutation | 0.354 | - | - | - | - | - |
| | Retrieved | 0.351 | **85.9** | **19.6** | 1.87 | - | - |
| Uncond | ProtGPT2 | 0.322 | 55.2 | 31.5 | 8.58 | 1.41 | **100** |
| | ProGen2 | 0.352 | 55.5 | 26.7 | 8.47 | 161.04 | **100** |
| Sequence | ZymCTRL | 0.375 | 62.5 | 23.0 | 4.12 | 0.78 | 99.2 |
| | NOS | 0.224 | 23.1 | 36.5 | 8.59 | **0.65** | **100** |
| Structure | LigandMPNN | 0.342 | 31.0 | 33.6 | 8.52 | 3.14 | 99.5 |
| | Ours | **0.380** | 62.8 | 20.8 | **1.74** | 0.90 | **100** |

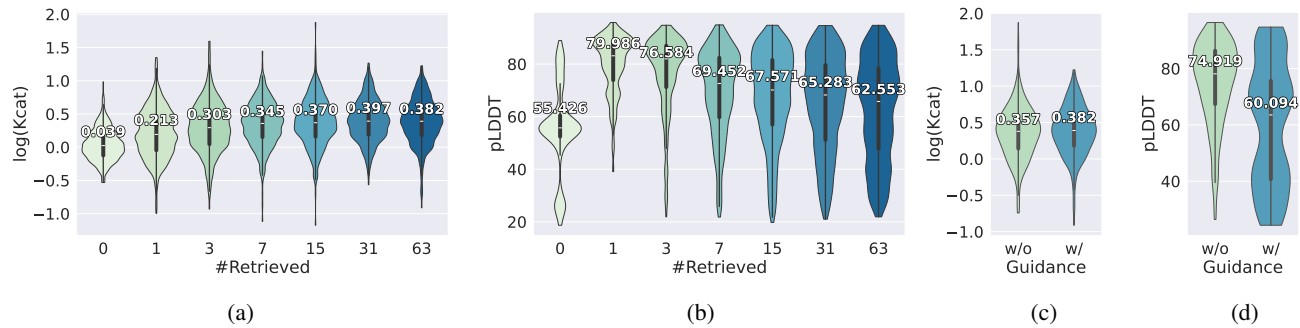

Figure 2: (a) and (b): The distribution of $k_{cat}$ value and pLDDT of our model with different numbers of retrieved enzymes. (c) and (d): Those of our model with or without discriminator guidance.

## 4.4. Retrieval Effectiveness

**Research question: Does the retrieval of enzymes contribute to enzyme generation?** We modified the number of retrieved enzymes and generated 10 enzymes for each of the 389 target substrates to evaluate the effectiveness of the retrieval method. Results are shown in Fig. 2(a-b).

▷ *Comparing generation with 0 and 1 retrieved protein in Fig. 2(a) and Fig. 2(b), it can be concluded that even a single retrieved enzyme is crucial to the generation of enzymes with catalytic capability and foldability.* It shows the effectiveness of the retrieval method.

▷ *Comparing generation with 1 or more retrieved proteins in Fig. 2(a) and Fig. 2(b), it can be concluded that retrieval enhances the generated enzymes' catalytic capability by a small concession of foldability.* Fig. 2(a) of $k_{cat}$ shows that the increase in retrieved sequences improves the performance in terms of catalyzing. In Fig. 2(b) of pLDDT, the foldability decreases with the increase of retrieved enzymes. The reason is that structure prediction examines the full sequence pattern with existing proteins. The retrieved proteins do not resemble each other in full sequence, making the derived generated sequence less similar to existing proteins. In fact, several short periods (enzymatic active site) in the retrieved sequences dominate the proteins' catalytic capability, which is different from foldability's requirement on the full sequence. Therefore, there's a trade-off between the enzyme's folding stability and catalytic capability. In fact, the trade-off has been reported in other literature (Vanella et al., 2024), which is the same case in our generated sequences. With more retrieved sequences, our model gives up sequence foldability for better catalytic performance.

## 4.5. Guidance Effectiveness

**Research question: Does the discriminator guidance contribute to enzyme generation?** We removed the discriminator in our model and generated 10 enzymes for each of the 389 target substrates to evaluate the guidance effectiveness. The results are shown in Fig. 2(c) and Fig. 2(d).

▷ *Fig. 2(c) shows the necessity of guidance to generate the enzymes with high $k_{cat}$.* Fig. 2(c) and Fig. 2(d) also suggest that our model performs the same trade-off in two circumstances with or without guidance. Comparing the $k_{cat}$ value of rule-based retrieved sequence in Table. 3 with w/o guidance column in Fig. 2(c), it can be seen that the generated enzymes' $k_{cat}$ is almost the same as the retrieved ones. The reason is that the generator only learns to generate sequences resembling the retrieved ones.

It is natural that adopting guidance decreases the foldability of generated enzymes. The discriminator guides the generator to output proteins with a high score, which has a different distribution from natural-like proteins. The pLDDT given by the structure prediction model suggests confidence, and it is low when the evaluated sequence is not very natural-like.

## 4.6. Case Study Targeting Methylphosphonate(1-)

**Research question: Why proteins generated by SENZ are predicted to have the better catalytic capability?** We perform docking between a substrate, methylphosphonate(1-) (Gama et al., 2019; von Arx et al., 2023), and generated enzymes with AutoDock-Vina[4] (Eberhardt et al., 2021) to closely examine the generated enzyme's structure and its interaction with the target substrate. The docking result is presented in Fig. 3.

▷ *From Fig. 3(f), it is evident that the enzyme generated by our model achieves the lowest AutoDock-Vina score, indicating the highest likelihood of binding between the molecule and the protein.* This result is likely due to our generated protein possessing more side chains that extend toward the substrate, resulting in a tighter binding. Although a favorable docking score does not necessarily ensure catalytic activity, it does demonstrate that our generated enzyme can effectively capture the substrate, which is a crucial prerequisite for the subsequent chemical reaction.

# 5. Related Work

**Unconditional protein generation.** Some research focuses on generating proteins that resemble natural ones. Within this scope, the protein language model-based sequence-only approaches include ProGen2 (Nijkamp et al., 2023), ProtGPT2 (Ferruz et al., 2022), and ESM-2 (Lin et al., 2023). These models are trained to predict masked amino acids in natural protein sequences, thus learning to generate proteins that mimic natural ones. The discrete diffusion models approach, aimed at this target, includes EvoDiff (Alamdari et al., 2023), which performs corruption and reconstruction on multiple sequence alignments (MSA). Generative adversarial networks (GAN) approaches, such as ProteinGAN (Repecka et al., 2021), use a discriminator to guide the generated protein to resemble natural ones, enabling the generation of natural-like enzymes when a template is provided. Structure-based methods include ProteinMPNN (Dauparas et al., 2022), which seeks to generate a protein sequence likely to fold into a given structure. These methods do not target external generation objectives or rely heavily on human-selected input templates to achieve specific functions.

**Conditioned protein generation.** Non-protein data is adopted to guide protein generation. ZymCTRL (Munsamy et al., 2022) uses an Enzyme Commission (EC) number as a prompt to generate enzymes categorized in the corresponding EC. ProGen (Madani et al., 2023) takes natural

---

[4] https://vina.scripps.edu

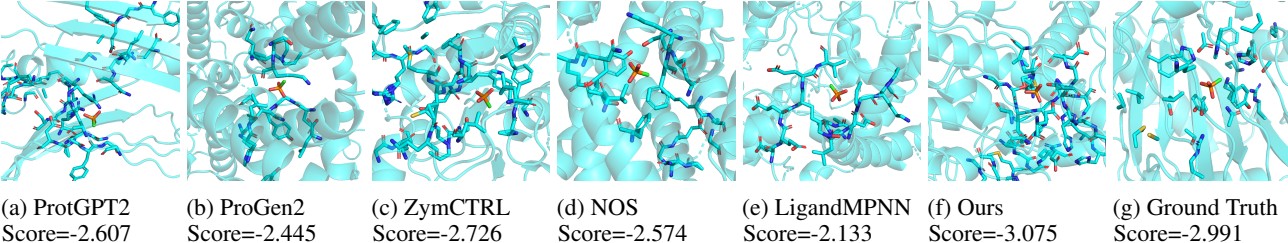

(a) ProtGPT2
Score=-2.607

(b) ProGen2
Score=-2.445

(c) ZymCTRL
Score=-2.726

(d) NOS
Score=-2.574

(e) LigandMPNN
Score=-2.133

(f) Ours
Score=-3.075

(g) Ground Truth
Score=-2.991

Figure 3: Docking result and the corresponding AutoDock-Vina scores of 6 neural network generated proteins for methylphosphonate(1-) and the ground truth. The molecule with 5 atoms in red, orange, and green is methylphosphonate(1-). The generated protein is in blue. Proteins' side chains within 5 Å to the substrate are shown. A lower score denotes a better binding position.

language protein labels to output corresponding protein sequence. ProCALM (Yang et al., 2024a) embeds taxonomy and EC number as conditions and use the condition embedding to generate related proteins. EnzyGen (Song et al., 2024b) integrates EC numbers and alignment-based identification of functionally important sites, thereby relying on both external label information and intrinsic sequence properties. LigandMPNN (Dauparas et al., 2025) and RFd-iffusionAA (Krishna et al., 2024) can recover a protein sequence and structure based on a binding molecule, which is derived from their prediction ability on the ligand-protein complex. GENZYME (Hua et al., 2024a) generates pocket structures and subsequently derives the corresponding sequences through pocket inverse folding, relying on structural training data and the performance of inverse folding models, particularly at functionally important sites.

**Protein guided protein generation.** Some research aims to generate new proteins that bind to a given protein. The sequence approach includes NOS (Gruver et al., 2023), which merges antibody and antigen in one sequence and uses the diffusion method to train a transformer, while some property prediction models can be used in sampling to make the generated protein tend to have certain properties. The structure approach includes AbDiffuser (Martinkus et al., 2023), which uses a SE(3) equivariant neural network to model residue-to-residue relations. The generation target and output protein are both in the same protein modality.

**Enzyme evaluation.** Enzyme evaluation models can help with enzyme design. ProSmith (Kroll et al., 2024) predicts protein-small molecule interactions. UniKP (Yu et al., 2023) predicts the $k_{cat}$ and $K_m$ value of enzyme and substrate. NeuralPLexer (Qiao et al., 2024) and AlphaFold 3 (Abramson et al., 2024) can predict the protein-ligand complex structures. Johnson et al. (2025) proposes comprehensive methods for evaluating neural network-generated enzymes but does not include metrics related to catalytic activity. Several enzyme-related datasets have been developed (Heid et al., 2023; Hua et al., 2024b; Yang et al., 2024b), though their primary focus on reaction prediction makes them less

suited for direct application to enzyme generation tasks.

**Retrieval method.** Retrieval-augmented generation (RAG) is widely adopted in a variety of domains (Shi et al., 2024; Shu et al., 2025; Zhang et al., 2025). Some research develops retrieval methods to help with generation or prediction. RetMol (Wang et al., 2023) retrieves molecules based on similarity and desired properties to refine molecules. MSA transformer (Rao et al., 2021) and AlphaFold 2 (Jumper et al., 2021) uses evolutionary-based MSA to enhance structure prediction accuracy. They retrieve proteins with proteins by sequence similarity only. RSA (Ma et al., 2024) retrieves proteins based on the embedding distance derived from sequences, thereby making it a method that also depends on the intrinsic properties of the protein sequences themselves.

## 6. Conclusion

In this paper, we have formally defined the task of zero-shot substrate-specified enzyme generation, wherein models are provided solely with a new target molecule and are required to output a protein sequence possessing catalytic capabilities specific to that molecule. To address this task, we introduce the **S**ubstrate-specified **enz**yme generator (SENZ), an RAG method. SENZ utilizes a single molecule as a query to retrieve enzymes based on their substrate similarity to the target, thereby enabling the retrieval of known proteins from new molecules. This retrieval strategy capitalizes on the functional similarity of enzymes as indicated by their substrates. To generate enzymes from the retrieved sequences, we employ multiple sequence alignment (MSA) on them and introduce a diffusion model generator guided by an enzyme-substrate classifier. This classifier guides the generated protein distribution for different substrates, serving as the objective for the generator during training. In experiments involving the generation of enzymes for real-world target molecules, evaluation functions assessed turnover rate and foldability together with other properties, demonstrating the superiority of our model in enzyme generation.

## Impact Statement

This paper presents a method for zero-shot substrate-specified enzyme generation, contributing to advancements in machine learning for protein design. Our approach has potential applications in biomaterial synthesis and industrial biocatalysis, with possible benefits for sustainable chemistry and pharmaceutical development. While generative models in biological design require careful validation and ethical oversight, our work is grounded in established biochemical principles and evaluation frameworks. We do not foresee immediate risks associated with misuse, as our method relies on controlled training data.

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

# A. Experiment Details

## A.1. Source of the 7 substrates used in the experiments and reason of selection

The seven substrates presented in Table 2 were chosen for their commonality and importance in enzymatic processes, highlighting their relevance as candidates for novel enzyme design.

Sepiapterin (Thöny et al., 2000). Reason for Designing Enzymes: Enhancing the efficiency and specificity of sepiapterin reductase can improve tetrahydrobiopterin (BH4) production, which is crucial for neurotransmitter synthesis and nitric oxide production.

Propylene Oxide (de Vries & Janssen, 2003). Reason for Designing Enzymes: Engineering epoxide hydrolases or monooxygenases can provide higher enantioselectivity and stability under industrial conditions for the production of chiral intermediates in pharmaceuticals.

Levoglucosan (Layton et al., 2011). Reason for Designing Enzymes: Developing specific glucosidases can enable efficient hydrolysis of levoglucosan into fermentable sugars, facilitating biofuel production from biomass pyrolysis products.

cGMP (Lucas et al., 2000). Reason for Designing Enzymes: Developing specific glucosidases can enable efficient hydrolysis of levoglucosan into fermentable sugars, facilitating biofuel production from biomass pyrolysis products.

L-Proline (L-Pro) (Tanner, 2008). Reason for Designing Enzymes: Engineering proline racemase or proline dehydrogenase can enhance the production of D-proline, a valuable chiral building block in pharmaceutical synthesis.

Pyridoxine (Bilski et al., 2000). Reason for Designing Enzymes: Designing enzymes for pyridoxine is important for enhancing its role in oxidative stress resistance by modulating its singlet oxygen quenching properties, which can be applied to improving fungal resilience, developing antioxidant therapies, and advancing fluorescence-based imaging techniques.

Leukotriene A4(1-) (Haeggstrom, 2000). Reason for Designing Enzymes: Developing selective leukotriene A4 hydrolase inhibitors can lead to new anti-inflammatory drugs with fewer side effects.

## A.2. More Details about baselines

ProGen2 (Nijkamp et al., 2023) and ProtGPT2 (Ferruz et al., 2022): We utilized the pre-trained weights for both models to generate sequences with a maximum length of 1024. These models serve as benchmarks for the capability of protein language models to generate sequences without specific functional guidance.

ZymCTRL (Munsamy et al., 2022): This model employs pre-trained weights and uses the Enzyme Commission (EC) number as a prompt for the autoregressive generation process. It is worth noting that the EC number provides more detailed information about enzymatic function compared to the substrate alone, offering this baseline an advantage in generating enzyme sequences for the given tasks.

NOS (Gruver et al., 2023): We trained NOS following the methodology of its original paper. The original NOS framework uses a discriminator to score the binding affinity between an antibody and an antigen (two protein sequences). We replaced the original discriminator with our enzyme-substrate probability scoring model in our adaptation. Furthermore, we replaced the target protein sequence input with the target substrate molecule. During inference, the NOS generator is updated iteratively for 10 steps using the test set input before sampling, following a discrete diffusion model for sequence generation, as described in the original paper. These adjustments allow NOS to generate enzymes in our setting while preserving its original generative framework.

LigandMPNN (Dauparas et al., 2025): This reverse folding model generates a protein sequence based on a protein-ligand complex structure. To adapt it for our task, we randomly generated protein sequences (length: 1024) and predicted their structures using ESMFold (Lin et al., 2023). Using RDKit, we generated the structure of the target substrate, and NeuralPLexer (Qiao et al., 2024) was employed to dock the substrate with the predicted protein structure, creating a complex structure. The resulting complex was then input into LigandMPNN for sequence redesign.

## A.3. Computation Resources

All the experiments are conducted within 200 GB memory, 2 Intel Xeon Gold 6426Y CPUs, and 4 NVIDIA 4090D GPUs with 24 GB memory each. All used data in the experiment requires storage of less than 500 GB.

The total training time of the models is less than 40 hours.

## B. Additional Related Work

**Graph Neural Networks for Molecular Modeling.** Graph Neural Networks (GNNs) have been extensively applied across various domains (Kipf & Welling, 2017; Veličković et al., 2018; Zhou et al., 2020a; 2021b; 2020b; Sun et al., 2022; Tan et al., 2023; Zhou et al., 2021a; Liu et al., 2023), and have shown particular promise in molecular representation learning. Guo et al. (2023) provide a comprehensive review of molecular graph modeling approaches, including 2D-based and 3D-based graph modeling methods. Fang et al. (2022) incorporate three-dimensional molecular geometry by constructing graphs informed by spatial structure and applying GNNs to capture geometric dependencies effectively. In the context of enhancing molecular representations, Tsubaki & Mizoguchi (2020) integrate atomic orbital features with graph convolutional networks (Kipf & Welling, 2017) to improve performance on downstream tasks. Beyond representation, GNNs have also been utilized for molecule generation; for instance, Therrien et al. (2025) employ gradient ascent on a trained GNN-based molecular property predictor to design novel molecules.

## C. Limitation

Currently, the implementation of our method can only deal with small molecule substrates. If users want to generate enzymes for polymer substrates like DNA, RNA, protein, or polysaccharides with our model, they have to derive the SMILES of the corresponding monomer or dimer manually for input.

