# OpenReview forum: "Retrieval Augmented Zero-Shot Enzyme Generation for Specified Substrate"
_ICML.cc/2025/Conference — ICML 2025 poster_

### Official Review · Reviewer_dRSM · 2025-03-13

**Overall Recommendation:** 2

**Summary:**

This paper introduces a novel method for de novo enzyme design using retrieval augmentation for the generative process. The core of this method is to take a given substrate for the enzyme, search in an enzyme database for protein sequences that are enzymes of similar substrates, align these protein sequences together, and provide these multiple sequence alignments to the generator model, which designs a sequence. Finally, a discriminator will predict how well the substrate and the generated sequence fit together. In addition, the authors construct a new enzyme-substrate database for training and retrieval.

**Claims And Evidence:**

This method generates enzyme protein sequences that it claims are superior to that of other methods. This is done by comparing the kcat predicted by UniKP of their generated sequences compared to other methods. However, this is not a reliable metric as UniKP is a simple regression model trained on protein language model embeddings, and although it is state of the art, it is not accurate enough to deem one model better than the other. This is doubly true in this case since the difference between the kcat’s of this model compared to other methods is small.

There are also claims made that do not seem to be true even if the UniKP model’s outputs are treated as ground truth. For example, the results in Figure 2. It is claimed in section 4.5 that “Fig. 2(c) shows the necessity of guidance to generate the enzymes with high kcat.” I do not agree with this interpretation of the figure’s data. In Fig. 2c, one can only see a slight improvement in the kcat, which is unremarkable when taking into account the distribution of the predicted kcat’s. On the other hand, the only significant result of the guidance seems to be decreasing the pLDDT by a large amount in Fig. 2d.

**Essential References Not Discussed:**

NA

**Experimental Designs Or Analyses:**

The experiments regarding foldability and designability of the generated sequences are conventional and have no issues. However, the central claims of this paper rely upon prediction of enzymatic kcat’s by UniKP. This is a computational model, relying upon Prot-T5 embeddings. Since these are only predictions, driven only by the sequence, it is hard to interpret the differences in kcat, especially minor ones such as those in this paper. In other words, the claims of this paper are too strong to merely be based on the kcat predictions of UniKP.

**Methods And Evaluation Criteria:**

It is difficult to evaluate whether generated enzymes achieve high catalytic efficiency in silico, which is the main issue of this paper. Even if this model is very performant, the evaluation criteria is not predictive or sensitive enough to make the claims that are underpin this paper.

On the other hand, I do like the use of AutoDock Vina for the docking results, as that is a structure-based method that has been widely validated. This could be interesting for further exploration, although the low confidence AlphaFold2 structures generated by this model currently will make docking unreliable as well.

**Other Comments Or Suggestions:**

NA

**Other Strengths And Weaknesses:**

See above

**Questions For Authors:**

Have the authors considered using AlphaFold3’s support for ligands to filter their predictions and score them against other models?

How does the sequence similarity of the binding pocket of the ligand look compared to the retrieved sequences from the database? Is the catalytic site stable or does the model make mutations here?

How sensitive are the model’s generations to the size of the database? Would a smaller database, particularly with less ligands available, make the outputs noticeably worse? How much more data is needed for the database to improve the model’s outputs?

The average log kcat is reported for Table 1. However, when designing enzymes de novo, it is likely that the sequences with max kcat will be tested. Could you also report max kcat of different methods for these tasks so that the shape of the distribution of the outputs becomes clear?

 Have the authors considered distillation of the UniKP model into the discriminator used here, letting the predictions of UniKP for the model’s designs train the discriminator further? Would this improve results?

**Relation To Broader Scientific Literature:**

This paper has designed and implemented a novel, creative, and highly promising method for protein sequence generation using retrieval augmentation. I think that this type of method will be very important for the advancement of the field. Furthermore, development of the substrate-enzyme database will accelerate enzyme research on its own.

**Theoretical Claims:**

The paper does not focus on formal theoretical claims.

---

> ### Author Rebuttal · Authors · 2025-04-01
>
> Dear Area Chair and Reviewers,
>
> We sincerely thank you for your thoughtful and thorough evaluation of our paper.
>
> # `UniKP for evaluation`
>
> Thanks for your comment. Using the predictive model for scoring is what we do in practice to filter better proteins. Before the wet lab experiment, proteins are scored by the predictive model, and proteins with a higher score are tested with higher priority. The advances in predicted scores suggest higher priority for wet lab experiments. We plan to test the designed enzymes in the wet lab, and the selection is based on the score.
>
> The evaluation of protein properties depends on machine learning model predictions, particularly when no traditional methods are available as alternatives. The direct *in silico* evaluation of catalytic capability is by adopting the predictive model, and it is also a necessary step for filtering before wet lab experiments. This process is also adopted in protein design tasks for different features with predictive models for different properties.
>
> # `Guidance for kcat improvement`
>
> In this manuscript, we want to show how far the $k_{cat}$ can achieve,  because a higher predicted $k_{cat}$ value makes designed proteins a higher priority for wet lab experiments. In real-world enzyme design, experts can decide the preference for predicted $k_{cat}$ or plddt, and then use a version of the model trained with or without guidance. We show the comprehensive effect of the module in facilitating experts' selection of model versions.
>
> # `AlphaFold3’s support for ligands to filter`
>
> Thank you for your suggestion. We will use more structure prediction models in evaluation in future versions of this manuscript.
>
> # ` Sequence similarity of the binding pocket in retrieved and the catalytic site`
>
> Thanks for your suggestion. The binding sites are usually not adjacent in sequence, so the sequence's similarity cannot be compared. The **binding sites** of an enzyme in the Uniprot annotation are usually discrete positions, which are not adjacent to each other in the protein sequence. Therefore, the sequence similarity cannot be compared based on several unconnected positions.
>
> There is no enough annotation related to every enzyme's **catalytic site**. Our dataset is created based on Rhea, which is an expert-curated knowledgebase of chemical and transport reactions of biological interest. However, there is no pocket or catalytic site annotation on the enzymes.
>
> # `Influence of database size`
>
> Thanks for your suggestion. We will use the full enzyme dataset in real-world new enzyme designing tasks. Of course, the number of substrates in the database decides the quality of retrieved proteins. Therefore, using the largest reliable enzyme knowledge base as the retrieval database is required in real-world tasks.
>
> # `Max kcat for Table 1`
>
> The max $k_{cat}$ value in Table 1 is in the table below.
>
> | Model        | Sepiap-terin | Propylene oxide | Levo- glucosan | cGMP  | L-Pro | Pyridoxine | leukotriene A4(1-) |
> | ------------ | ------------ | --------------- | -------------- | ----- | ----- | ---------- | ------------------ |
> | Ground Truth | 0.703        | 0.785           | 0.736          | 0.288 | 0.159 | 0.573      | 0.676              |
> | Random       | 0.050        | 0.303           | 0.734          | 0.072 | 0.311 | 0.616      | 0.056              |
> | Mutation     | 0.898        | 0.838           | 0.837          | 0.452 | 0.155 | 0.527      | 0.666              |
> | Retrieved    | 0.409        | 0.955           | 0.981          | 0.401 | 0.156 | 0.639      | 0.856              |
> | ProtGPT2     | 0.816        | 0.910           | 1.028          | 0.621 | 0.724 | 0.769      | 0.555              |
> | ProGen2      | 1.244        | 0.771           | 0.965          | 0.833 | 0.789 | 0.810      | 0.800              |
> | ZymCTRL      | 0.004        | 0.604           | 0.774          | 0.348 | 0.454 | 0.728      | 0.538              |
> | NOS          | 0.349        | 0.590           | 0.488          | 0.341 | 0.412 | 0.711      | 0.447              |
> | LigandMPNN   | 0.653        | 0.778           | 1.174          | 0.251 | 0.497 | 0.786      | 0.976              |
> | Ours         | 1.102        | 1.117           | 1.065          | 0.642 | 0.909 | 0.934      | 1.470              |
>
> # `Distillation of the UniKP into the discriminator`
>
> Thanks for your suggestion. In real-world tasks, we will consider utilizing other predictive models, including UniKP, as the discriminator as long as it is end-to-end and preserves the gradient. We will also try to use predictions for further rounds of training. The reason we do not adopt it in the current manuscript is that the evaluation relies on UniKP, which should not be seen by modules in training and generation.

---

### Official Review · Reviewer_Xwvh · 2025-03-14

**Overall Recommendation:** 3

**Summary:**

The paper introduces SENZ, a substrate-specified enzyme generator, a RAG-based method to retrieve known enzymes and generate new enzymes based on a substrate. The authors define the task as generating a protein that serves as an enzyme for a given small molecule target. As a first step, the authors design a retrieval algorithm that retrieves a protein for a given query molecule based on catalyzing properties. Then a discrete diffusion model generates new enzymes conditioned on the retrieved enzymes and substrate.

## Update After Rebuttal

I thank the authors for addressing my review. I have decided to stay with my score of weak accept.

**Claims And Evidence:**

* The authors claim that SENZ can generate enzymes for a particular substrate.
* The retrieval quality is not measured.

The claim of the model being superior to any others is slightly not convincing. There’s essentially only 3 comparisons across the models: The average magnitude of the turnover enzyme number over 7 tasks, the average property scores, and the docking score of one target. The second one is fine. The first one only reports on the average performance of each model’s generation, but doesn't include the generation performance comparison, which I think is equally important. The third one seems to be cherry-picking as there’s only one target being compared. I have detailed what I’d like to see in the question part below.

**Essential References Not Discussed:**

N/A

**Experimental Designs Or Analyses:**

See Evaluation Criteria.

**Methods And Evaluation Criteria:**

Section 4.2: The authors assess the capability of SENZ to generate new proteins by generating new enzymes given a substrate and computing the turnover number
The values in Table 1 are a little strange. Random enzymes are significantly worse than unconditionally generated enzymes, but they are also randomly generated. Standard deviations are not provided along with the mean scores for each task, but possibly more than 10 sequences are needed to get valid evaluation results
Shouldn’t the retrieved enzymes match the ground truth proteins? Is the retrieval from a different set of enzymes?
Line 297: How are unconditionally generated enzymes different from natural enzymes?
In general, the proposed metric is not a strong indicator of the efficacy of the metric.

**Other Comments Or Suggestions:**

Line 143 Col 1: Unclear what “(with an identity exceeding 30%)” means
The notation in equation 6 is non-standard and a bit confusing. In equation 5, enzymes are presented as x but then enzymes are represented as P(m_i). Is P(m_i) a set of enzymes as there may be multiple protein matches given a substrate?
Line [157-163 Col 2: ]The following paragraph is difficult to parse.

**Other Strengths And Weaknesses:**

The retrieval component of the pipeline is very similar to fuzzy search algorithms and semantic search algorithms and is not particularly novel.
Substructure similarity is measured with Tanimoto distance of Morgan fingerprints which does not capture 3D information.

**Questions For Authors:**

How is the diversity and novelty of your model’s generation? For example, for a given substrate generated enzymes, how different are they, and how many are previously not seen in the database or from the testing set?
For the generation results you have in Experiment 4.1, what’s the average docking score for each to its target?

**Relation To Broader Scientific Literature:**

N/A

**Theoretical Claims:**

N/A

---

> ### Author Rebuttal · Authors · 2025-04-01
>
> Dear Area Chair and Reviewers,
>
> We sincerely thank you for your thoughtful evaluation.
>
> # `Retrieval quality`
>
> The quality of retrieved enzymes is reflected in the baseline method named '**Retrieved**' in Table 2. That row is as follows:
>
> | Method    | $k_{cat}$ | pLDDT |
> | --------- | --------- | ----- |
> | Retrieved | 0.351     | 85.9  |
>
> # `Random versus unconditionally generated enzymes`
>
> 1. 'Random' are randomly generated amino acid sequences. They are not randomly selected enzymes.
> 2. Unconditional generated proteins reflect the distribution of the training set of the baseline method. These models are trained to generate natural-like proteins.
>
> The comparison reflects the features of $k_{cat}$ prediction model in evaluation. A more natural-like protein obtains a higher score than a pure random amino acid sequence.
>
> # `Retrieved enzymes versus ground truth`
>
> The retrieval is from a different set which contains no ground truth enzymes. The retrieval module can never see the ground truth proteins.
>
> # `Line 297: Natural versus unconditionally generated enzymes`
>
> 1. Natural enzymes are ground truth, which are the enzymes for target substrates.
> 2. Unconditionally generated enzymes are generated by baseline models. They do not exist in nature.
>
> # `Novelty of retrieval and substrate 3D information`
>
> Our novelty is searching similar substrates and using their enzymes as reference for the generation model, not the metric about molecule similarity. We use a well-recognized method to calculate the similarity between molecules in the retrieval database. Morgan Fingerprint (MF) is an easy and recognized method to map molecules into the same vector space as representation. Tanimoto distance is designed to measure the similarity between vectors like MF.
>
> MF represents structure without relying on 3D coordinates. It encodes atom environments in a molecule, capturing local atomic neighborhoods. It can indicate whether certain functional groups or motifs are present in a molecule. It allows the comparison of molecules based on shared substructures.
>
> # `Identity exceeding 30%`
>
> It means using `mmseqs2` with `--min-seq-id 0.3`. In Line 143, it means any two protein sequences from different subsets (training set, validation set, or testing set) have a maximum of 30% position overlap. This restriction is to avoid data leakage in protein sequence dataset split. Proteins are not relevant under the threshold.
>
> # `Eq 5,6 and Line [157-163 Col 2]`
>
> Notation for protein:
>
> 1. $x$ is a single protein.
> 2. $\mathbb{P}^{(m)}$ is a set of proteins related to molecule $m$. If the molecule is in the retrieval database, these proteins are the enzymes of the molecule; If the molecule is not in the retrieval database,  these proteins are the enzymes of similar molecules.
>
> Equation 6 is a definition relying on equation 5: if the target molecule is not in the retrieval dataset, the proteins for some molecules in the dataset will be collected and serve as the proteins for the target molecule.
>
> Line [157-163 Col 2: ] is the process of retrieving proteins for a target molecule.
>
> 1. Calculate the similarity between the target molecule and molecules in the database.
> 2. Order molecules in the database descending by similarity.
> 3. Get all the enzymes of each molecule in database in order until a desired amount is obtained.
> 4. If a molecule has too many enzymes, only part of them is selected. It is to prevent all retrieved proteins from being enzymes of a single molecule.
>
> # `Diversity and novelty`
>
> 1. The **diversity** of generated enzymes for the same substrate.
>
>    It is measured by the metric **#clusters**. Enzymes for the same target substrate are generated at first. Then they are clustered by a threshold of 30% identity, which means every two proteins from different clusters do not have an overlap of 30%. The number of clusters is recorded. Ground truth enzymes of the same substrate are also clustered, and the number of clusters is obtained. The absolute difference between clusters is the resemblance of diversity to natural distribution. If the generated enzymes have approximately the same number of clusters as natural ones, they have a natural distribution in terms of diversity.
>
> 2. The **novelty** of generated enzymes.
>
>    It is measured by the metric **BLASTp**. It can be regarded as the distance from a protein to the nearest in all other natural proteins. It is the identity of the generated enzyme with the nearest different known protein, which is obtained by BLASTp2 in the SwissProt database. We also get this value of natural enzymes, and the absolute difference between it and of generated proteins is calculated. If the distance of generated enzymes is similar to that of natural ones, they have a natural distribution in novelty.
>
>
> # `Average docking score `
>
> We will calculate the docking score in the next version of the manuscript. Runnig docking needs to manually adjust the protein structure file with the substrates'.

---

### Official Review · Reviewer_aDZN · 2025-03-14

**Overall Recommendation:** 2

**Summary:**

This paper presents a retrieval-augmented approach for enzyme sequence generation, conditioned on substrates. The authors introduce a novel benchmark designed to address the zero-shot setting, which is aligned with downstream generation tasks. The proposed method utilizes a diffusion model to iteratively generate enzyme sequences, conditioned on multiple sequence alignment results and the target substrate. Empirical evaluations demonstrate that their method outperforms existing baselines in terms of generation quality.

**Claims And Evidence:**

The authors review existing protein generation methods and assert that "existing work on conditional generation cannot be repurposed directly to generate desired enzymes that catalyze specific substrates represented as small molecules" (lines 42-45). However, this claim overlooks several existing approaches that address enzyme generation with a focus on catalysis, such as ProCALM[1], GENZYME[2], and EnzyGEN[3]. Additionally, in the context of enzyme sequence generation, there are notable datasets and benchmarks that should be considered, including ReactZyme[4], CARE[5], EnzymeMap[6], and EnzyBench[3].

**Reference:**

[1] Yang J, Bhatnagar A, Ruffolo J A, et al. Conditional enzyme generation using protein language models with adapters[J]. arXiv preprint arXiv:2410.03634, 2024.

[2] Hua C, Lu J, Liu Y, et al. Reaction-conditioned De Novo Enzyme Design with GENzyme[J]. arXiv preprint arXiv:2411.16694, 2024.

[3] Song Z, Zhao Y, Shi W, et al. Generative enzyme design guided by functionally important sites and small-molecule substrates[J]. arXiv preprint arXiv:2405.08205, 2024.

[4] Hua C, Zhong B, Luan S, et al. Reactzyme: A benchmark for enzyme-reaction prediction[J]. Advances in Neural Information Processing Systems, 2025, 37: 26415-26442.

[5] Yang J, Mora A, Liu S, et al. CARE: a Benchmark Suite for the Classification and Retrieval of Enzymes[J]. arXiv preprint arXiv:2406.15669, 2024.

[6] Heid E, Probst D, Green W H, et al. EnzymeMap: curation, validation and data-driven prediction of enzymatic reactions[J]. Chemical Science, 2023, 14(48): 14229-14242.

**Essential References Not Discussed:**

Please see the section of **Claims And Evidence**.

**Experimental Designs Or Analyses:**

I have some concerns regarding the validity of the $k_{cat}$ results. The authors claim that their model performs better than the baselines, and even outperforms the ground truth. However, in the case study, the reported Vina score is -3.075, which indicates that binding with the target substrate is unlikely. How can the authors reconcile these discrepancies in their results? Additionally, they should compare their docking results with the ground truth for a more reliable assessment.

The authors use pLDDT to evaluate sequence validity, which is an appropriate initial step. However, once the validity of the generated sequences is confirmed, further evaluations should follow, such as computing $k_{cat}$ and binding affinity. It would be helpful if the authors reported the $k_{cat}$ for the proportion of valid sequences as part of this process.

The proposed approach retrieves sequences based on substrate similarity rather than protein similarity, which distinguishes it from other methods. Can the authors provide evidence to demonstrate the significance of this design choice?

Finally, the authors only assess the influence of substrate guidance as part of the contrastive learning loss. However, the role of the concatenation operation for the generator remains unclear. Could the authors elaborate on the rationale behind this design choice and its impact on model performance?

**Methods And Evaluation Criteria:**

In the "Problem definition" part, the authors formalize the generative function as $\textbf{x}=G(\textbf{m},s_{EC},\textbf{C}_m,\textbf{C}_x)$, where they consider $\textbf{m}$ as a single condition during training. However, this raises the question of whether their approach effectively addresses the problem outlined in the Introduction, namely that "the catalytic capability of an enzyme is not solely determined by how it structurally interacts with the substrate molecule." Does the current setting fully capture the complexity of enzyme catalysis beyond just the structural interactions?

The authors also use $k_{cat}$ as an evaluation criterion, which is derived from a predictive model known to have significant limitations. This raises concerns about the reliability of the results. The authors should provide a justification for the reliability of the selected predictive model. Additionally, since the model is designed to generate enzyme sequences that can bind to the target substrates, it would be important to also compute the binding affinity as part of the evaluation.

Please clearly indicate what novel contributions come from the authors versus have been taken from other papers and combined. For example, it seems that most of the generative model is based on diffusion model and not introduced by the authors. Is the approach for substrate-conditioning developed by you or by others? The substrate-indexed retrieval method for example (to the best of my knowledge) is a novel contribution of the authors.

As discussed in Section 2, there are several enzyme-substrate datasets available. What differentiates your dataset from others in terms of quality and coverage? For example, EnzyBench contains over 100,000 enzyme-substrate pairs, which is a large and diverse dataset. Given that sequences are typically much easier to gather than structures for proteins or enzymes, can the authors ensure that their model is capable of learning meaningful patterns with a more limited dataset?

Additionally, the authors claim in line 50 that learning the interactions between substrates and enzymes is challenging. However, their method simply concatenates the embedding of the target substrate with the noisy enzyme sequence during training. Does this approach effectively address the complexity of learning substrate-enzyme interactions, or does it oversimplify the problem?

**Other Comments Or Suggestions:**

They should provide more details for the selected seven tasks.

**Other Strengths And Weaknesses:**

None

**Questions For Authors:**

None.

**Relation To Broader Scientific Literature:**

This paper introduces a method for zero-shot substrate-specific enzyme generation, making a contribution to the field of machine learning for protein design. The proposed approach has potential applications in biomaterial synthesis and industrial biocatalysis, with possible implications for sustainable chemistry and pharmaceutical development.

**Theoretical Claims:**

There is no theoretical proof.

---

> ### Author Rebuttal · Authors · 2025-04-01
>
> Dear Area Chair and Reviewers,
>
> We sincerely thank you for your thoughtful and thorough evaluation of our paper.
>
> # `Mentioned reference`
>
> We will include these six pieces of literature in the related work section in the final version of this paper since modification to the manuscript is not allowed currently. We plan to test our model on the mentioned dataset.
>
> # `Complexity of catalysis`
>
> Our approach to capture the complexity of enzyme catalysis beyond just the structural interactions is **by generation based on reference enzymes**. Catalytic capability is not fully decided by binding affinity or docking position but also by some unclear features that cannot modeled directly. The reason to use similar molecules' enzymes as references is these enzymes are known to function for similar molecules and thus contain necessary features even if the feature itself is unclear. The discriminator guidance provides clues to necessary features by serving as a loss.
>
> # `Predictive model and Binding affinity`
>
> The reliability of the predictive model can be checked from its original paper. In our paper, the rule-based method generated enzymes' predicted $k_{cat}$ is reasonable in terms of order, since the totally random generated protein << retrieved proteins $\approx$ mutation of ground truth < ground truth. The rule-based baselines are to evaluate if the predictive model can predict values in reasonable order.
>
> To reflect binding affinity, Vina score is provided in the case study. We are still implementing massive automated evaluation by AutoDock-Vina, since there are frequent cases unsuitable for the input check.
>
> # `Novelty`
>
> The approach of expressing substrate conditions to the model is novel, which includes two novel parts.
>
> 1. **Retrieval based on substrates**
>
>    The idea of using a substrate to find related enzymes and using related emzyes to facilitate new enzyme generation is novel. The approach to align related enzymes and serve as the base of generation is novel. This approach makes zero-shot generation practical.
>
> 2. **Adopt discriminator in training**
>
>    The approach of using a discriminative mode to describe the generation target of catalytic capability is novel.
>
> # `Dataset`
>
> Our dataset is created based on Rhea, which is an expert-curated knowledgebase of chemical and transport reactions of biological interest - and the standard for enzyme and transporter annotation in UniProtKB. Our dataset inherits the quality and coverage from Rhea.
>
>  To control the quality, we carefully designed the **dataset split**.
>
> 1. Sequence identity threshold
>
>    EnzyBench partitioned training, validation, and testing sets based on a sequence identity threshold of 50%, but ours is 30%.
>
> 2. Distribution of enzymes in split
>
>    Unlike EnzyBench split enzymes based on their third-level categories, we split the enzymes by substrates. We guarantee that no two enzymes in different splits share the same substrate.
>
> There are 34982 entries, so it cannot be regarded as limited, given the fact that high-quality annotated enzyme data is precious.
>
> # ` Substrate-enzyme interaction complexity`
>
> Our model does not aim to model the substrate-enzyme interactions directly, but to use retrieved reference enzymes to facilitate generation. We use the target substrate to retrieve enzymes, which contain the desired enzyme's properties. The model does not need to model substrate-enzyme interactions directly but needs to learn generation from references.
>
> # `Vina score`
>
> We provide the **docking results with the ground truth** here. The Vina score of the ground truth enzyme is **-3.036**, which is very close to our generated enzyme. It means that our generated enzymes bind the substrate slightly better than the ground truth with no discrepancy. We will add a subfigure in Figure 3.
>
> About the value -3.075 and -3.036. The ligand only has five atoms and thus is small. It is hard for a small ligand to get a high score.
>
> # $k_{cat}$`  for valid sequences`
>
> Thanks for your suggestion. When setting plddt threshold as 70%, our model-generated protein above the threshold has an average predicted $k_{cat}$ of **0.362**.
>
> # `The significance of retrieval`
>
> Thank you for identifying our novelty. This design aims to provide indirect substrate information in the format of protein sequences. Substrate-based retrieval is adopted for two significant reasons.
>
> 1. Without any protein as input, there is no anchor protein sequence to search for similar sequences.
> 2. With only a target substrate as input, the retrieval method has to use this molecule as the key.
>
> # `Substrate concatenation`
>
> It provides direct information on the target substrate by serving as a single token prompt persistent in the diffusion process. The sequence to generate has known positions from two dimensions then:
>
> 1. The rows of aligned sequences.
> 2. The column of the target molecule's embedding.
>
> # `7 tasks`
>
>  Please refer to the response to Reviewer MkMZ.

---

### Official Review · Reviewer_MkMZ · 2025-03-21

**Overall Recommendation:** 4

**Summary:**

This manuscript proposes the SENZ method for zero-shot substrate-specified enzyme generation. Its key points include: (1) defining the task and constructing a substrate-enzyme dataset; (2) retrieving relevant enzymes based on substrate similarity; (3) generating new enzymes via a diffusion model guided by a classifier for optimization; and (4) validating on diverse substrates with superior performance over existing methods.

**Claims And Evidence:**

Yes

**Essential References Not Discussed:**

No

**Experimental Designs Or Analyses:**

Yes

**Methods And Evaluation Criteria:**

Yes

**Other Comments Or Suggestions:**

It is recommended that when the "Uncond" class methods are first introduced in the experimental section, a brief explanation of how unconditional generation models (such as ProtGPT2 and ProGen2) are utilized to generate enzyme sequences for specific substrates should be provided.

**Other Strengths And Weaknesses:**

This manuscript has several notable strengths:

1.The SENZ method achieves zero-shot substrate-specified enzyme generation, innovatively proposing a way to generate novel enzymes for specific target substrates without direct supervision data.
2.The method integrates retrieval-augmented generation technology, addressing the lack of positive samples in zero-shot scenarios by retrieving enzymes with substrates structurally similar to the target as prompting signals.
3.The generated enzymes are comprehensively evaluated across multiple dimensions, including catalytic capability (kcat), foldability (pLDDT), and similarity to known enzymes (BLASTp), with experimental results demonstrating the effectiveness of the method.

However, the manuscript has the following issues:

1.The 7 substrates used in the experiments (Sepiapterin, Propylene oxide, Levo-glucosan, cGMP, L-Pro, Pyridoxine, leukotriene A4(1-)) listed in Table 1—where do they originate from? Please specify the source and explain why these particular substrates were selected for the experiments.

2. It is suggested that the authors consider removing structural elements such as signal peptides and transmembrane domains, which may affect expression and activity. This would enhance the feasibility and efficiency of the generated enzymes in practical applications, particularly in in vitro expression systems.

**Questions For Authors:**

Same to Other Strengths And Weaknesses

**Relation To Broader Scientific Literature:**

All good

**Theoretical Claims:**

Yes

---

> ### Author Rebuttal · Authors · 2025-04-01
>
> Dear Area Chair and Reviewers,
>
> We sincerely thank you for your thoughtful and thorough evaluation of our paper. Below are our detailed responses to your comments:
>
> ## **`Source of the 7 substrates used in the experiments and reason of selection`**
>
> Thanks for your comment. We selected these substrates because they are common and important in enzymatic reactions. Then, it is worthy to design new enzymes for these substrates.
>
> 1. Sepiapterin [1]
>
>    **Reason for Designing Enzymes**: Enhancing the efficiency and specificity of sepiapterin reductase can improve tetrahydrobiopterin (BH4) production, which is crucial for neurotransmitter synthesis and nitric oxide production. [1]
>
>    [1] Thöny B, Auerbach G, Blau N. Tetrahydrobiopterin biosynthesis, regeneration and functions. Biochem J. 2000 Apr 1;347 Pt 1(Pt 1):1-16.
>
> 2. Propylene Oxide [2]
>
>    **Reason for Designing Enzymes**: Engineering epoxide hydrolases or monooxygenases can provide higher enantioselectivity and stability under industrial conditions for the production of chiral intermediates in pharmaceuticals. [2]
>
>    [2] Erik J de Vries, Dick B Janssen, Biocatalytic conversion of epoxides. Current Opinion in Biotechnology. Volume 14, Issue 4, 2003, Pages 414-420.
>
> 3. Levoglucosan [3]
>
>    **Reason for Designing Enzymes**: Developing specific glucosidases can enable efficient hydrolysis of levoglucosan into fermentable sugars, facilitating biofuel production from biomass pyrolysis products. [3]
>
>    [3] Donovan S. Layton, Avanthi Ajjarapu, Dong Won Choi, Laura R. Jarboe. Engineering ethanologenic Escherichia coli for levoglucosan utilization. Bioresource Technology,
>    Volume 102, Issue 17, 2011, Pages 8318-8322.
>
> 4. cGMP [4]
>
>    **Reason for Designing Enzymes**: Developing specific glucosidases can enable efficient hydrolysis of levoglucosan into fermentable sugars, facilitating biofuel production from biomass pyrolysis products. [4]
>
>    [4] Lucas KA, Pitari GM, Kazerounian S, Ruiz-Stewart I, Park J, Schulz S, Chepenik KP, Waldman SA. Guanylyl cyclases and signaling by cyclic GMP. Pharmacol Rev. 2000 Sep;52(3):375-414.
>
> 5. L-Proline (L-Pro) [5]
>
>    **Reason for Designing Enzymes**: Engineering proline racemase or proline dehydrogenase can enhance the production of D-proline, a valuable chiral building block in pharmaceutical synthesis. [5]
>
>    [5] Tanner JJ. Structural biology of proline catabolism. Amino Acids. 2008 Nov;35(4):719-30.
>
> 6. Pyridoxine [6]
>
>    **Reason for Designing Enzymes**:  Designing enzymes for pyridoxine is important for enhancing its role in oxidative stress resistance by modulating its singlet oxygen quenching properties, which can be applied to improving fungal resilience, developing antioxidant therapies, and advancing fluorescence-based imaging techniques. [6]
>
>    [6] Bilski P, Li MY, Ehrenshaft M, Daub ME, Chignell CF. Vitamin B6 (pyridoxine) and its derivatives are efficient singlet oxygen quenchers and potential fungal antioxidants. Photochem Photobiol. 2000 Feb;71(2):129-34.
>
> 7. leukotriene A4(1-) [7]
>
>    **Reason for Designing Enzymes**: Developing selective leukotriene A4 hydrolase inhibitors can lead to new anti-inflammatory drugs with fewer side effects.
>
>    [7] Haeggström JZ. Structure, function, and regulation of leukotriene A4 hydrolase. Am J Respir Crit Care Med. 2000 Feb;161(2 Pt 2):S25-31. doi: 10.1164/ajrccm.161.supplement_1.ltta-6.
>
> ## **`To remove structural elements such as signal peptides and transmembrane domains`**
>
> Thanks for your suggestion. This process can provide the model with clearer sequence patterns by eliminating undetermined sections. In the future version, we will test the performance after removing signal peptides and transmembrane domains in the whole dataset.
>
> ## **`Detail of unconditional generation baselines`**
>
> ProGen2 and ProtGPT2: We utilized the pre-trained weights for both ProGen2 and ProtGPT2 to directly generate sequences with a maximum length of 1024. No information related to target substrates is provided to either model. These models serve as baselines for the capability of protein language models to generate sequences without specific functional guidance.

---

### Decision · Program_Chairs · 2025-05-01

**Decision:**

Accept (poster)

**Comment:**

The paper proposes SENZ, a novel retrieval-augmented generative framework for zero-shot enzyme design, where the goal is to generate enzymes that catalyze substrates for which no known enzyme exists. The method retrieves structurally similar substrates and their associated enzymes from a database, aligns them, and uses a discrete diffusion model to generate new enzyme sequences.

Pros:
- Several authors recognized the novelty of the approach. The idea of using retrieval augmentaion in enzyme design with diffusion models is novel.
- Promising performance across several metrics
- Diversity, novelty and binding score measurements are commonly used and trusted in the field.

Cons:
- Several key publications and datasets were not included.
- k_cat predicted by UniKP is used for evaluation, which is not robust.
- retriveal quality is not studied

While there are valid critisms on the evaluation protocol in this submission, given the novelty of this submission, I believe the merits of this submission outweighs its weaknesses.